# A systematic review and meta-analysis of antimicrobial resistance knowledge, attitudes, and practices: Current evidence to build a strong national antimicrobial drug resistance narrative in Ethiopia

**Beshada Zerfu Woldegeorgis**[ID][1]*, **Amene Abebe Kerbo**[2], **Mohammed Suleiman Obsa**[3], **Taklu Marama Mokonnon**[4]

1 School of Medicine, Wolaita Sodo University, Wolaita Sodo, Ethiopia, 2 School of Public Health, Wolaita Sodo University, Wolaita Sodo, Ethiopia, 3 Department of Anesthesia, Arsi University, Assela, Ethiopia, 4 School of Midwifery, Wolaita Sodo University, Wolaita Sodo, Ethiopia

* beshadazerfu@gmail.com

## Abstract

Antimicrobial resistance (AMR) is a silent pandemic that has claimed millions of lives, and resulted in long-term disabilities, limited treatment options, and high economic costs associated with the healthcare burden. Given the rising prevalence of AMR, which is expected to pose a challenge to current empirical antibiotic treatment strategies, we sought to summarize the available data on knowledge, attitudes, and practices regarding AMR in Ethiopia. Articles were searched in international electronic databases. Microsoft Excel spreadsheet and STATA software version 16 were used for data extraction and analysis, respectively. The Preferred Reporting Items for Systematic Reviews and Meta-Analysis 2020 checklist was followed. The methodological quality of the studies included was assessed by the Joana Briggs Institute critical appraisal checklists. The random-effect meta-analysis model was used to estimate Der Simonian-Laird's pooled effect. Statistical heterogeneity of the meta-analysis was checked through Higgins and Thompson's $I^2$ statistics and Cochran's Q test. Publication bias was investigated by funnel plots, and the regression-based test of Egger for small study effects with a $P$ value < 0.05 was considered to indicate potential reporting bias. In addition, sensitivity and subgroup meta-analyses were performed. Fourteen studies with a total of 4476 participants met the inclusion criteria. Overall, the pooled prevalence of good AMR knowledge was 51.53% [(95% confidence interval (CI): 37.85, 65.21), $I^2$ = 99.0%, P <0.001]. The pooled prevalence of favorable attitudes and good practices were 63.43% [(95% CI: 42.66, 84.20), $I^2$ = 99.6, P <0.001], and 48.85% [(95% CI: 38.68, 59.01), $I^2$ = 93.1, P <0.001] respectively. In conclusion, there is a significant knowledge and practice gap on AMR among the general public, patients, and livestock producers. As a result, we call for greater educational interventions to raise awareness and build a strong national AMR narrative.

**Data Availability Statement:** All relevant data are within the manuscript and its Supporting Information files.

**Funding:** The author(s) received no specific funding for this work.

**Competing interests:** The authors have declared that no competing interests exist

**Abbreviations:** AMR, Antimicrobial resistance; CI, Confidence interval; JBI, Joana Briggs Institute; KAP, Knowledge, attitudes and practices; PRISMA, Preferred Reporting Items for Systematic Reviews and Meta-Analyses.

## Introduction

Antimicrobial resistance (AMR) is the ability of microorganisms to become increasingly resistant to an antimicrobial drug to which they were previously susceptible, resulting in infections persisting in the body and increasing the risk of spreading to the human and animal population through food, water, and the environment [1]. According to the Food and Agriculture Organization's AMR action plans (2016–2020), AMR affects every sector and actor, regardless of economic status or geographic location [2]. Furthermore, AMR is a serious public health threat and has resulted in negative outcomes such as severe illnesses, longer hospital stays, long-lasting disabilities, increased healthcare costs, an overburdened public health system, higher costs for second-line drugs, treatment failures, and increased mortality rates [3–5].

Rising levels of AMR will impede progress toward many of the sustainable development goals, particularly those focusing on health and well-being, poverty reduction, food security, environmental well-being, and socioeconomic growth and development, according to antimicrobial resistance and the United Nations sustainable development cooperation framework guidance for United Nations country teams [6]. Bodies of evidence suggest that the inappropriate uses of antimicrobials in humans, animals, and plants; inadequate sanitation, hygiene, biosecurity, and infection prevention and control measures in health care settings, communities, farms, and food production systems; a lack of equitable access to affordable and quality-assured antimicrobials, vaccines, and diagnostics; the natural evolution, mutation, and transmission of resistant genes through horizontal gene transfer; and the failure to develop new antimicrobials consistently are all factors that have expedited the emergence and spread of AMR [6–10]. In 2016, the United Kingdom government commissioned an AMR review, which estimated that drug-resistant pathogens kill 700,000 people worldwide each year and that AMR could kill up to 10 million people per year [11]. Besides, the World Bank estimated that if AMR is not addressed, by 2050 the global economy may have lost nearly 4% of its annual gross domestic product, with the losses being even greater in low- and middle-income countries [12]. An estimated 4.95 million deaths were associated with bacterial AMR in 2019, according to a systematic analysis of the global burden of bacterial AMR, with Sub-Saharan Africa having the highest all-age death rate attributable to and associated with AMR at 23.7 deaths per 100,000 people [13]. A higher burden of AMR has been reported in countries where people have inadequate knowledge about it [14]. According to a research report by Machowska et al., a lack of adequate knowledge and awareness was a major driver of AMR in Europe [15]. Furthermore, a Pakistani study found that only 56.3% of the general population was aware that they could help reduce AMR [16]. Similarly, a nationwide online survey in Japan found that a significant number of people lacked adequate knowledge about AMR [17].

Despite efforts to combat AMR, barriers remain in Ethiopia, according to a baseline national survey on antimicrobial use, resistance, and containment conducted by the Drug Administration and Control Authority [18]. Furthermore, Berhe et al. found that pathogens causing a variety of diseases were resistant to 30–85% of important antimicrobial agents listed in the national standard treatment guideline [19]. This was expected to pose a challenge to current empirical antibiotic treatment strategies given the rising prevalence of AMR in Ethiopia.

To the best of our knowledge, there has been no systematic review and meta-analysis conducted in Ethiopia regarding AMR. Therefore, we sought to summarize available data on good knowledge, favorable attitudes, and good practices toward AMR. Identifying gaps and improving the public's knowledge, attitude, and practices about AMR, as well as those of patients, livestock producers, and healthcare workers is critical to putting an end to the persistent trends of increasing AMR. Because AMR has become a cross-cutting public health issue, our study results will shed a green light on the importance of a One Health approach for understanding

and mitigating AMR, and thus the key findings of this review will influence policymakers, researchers, and stakeholders to establish and strengthen inter-sectorial collaboration to optimize AMR perception and practices.

## Materials and methods

### Study protocol registration, and reporting

We have undertaken this systematic review and meta-analysis to estimate the pooled prevalence of good knowledge, favorable attitudes, and good practices toward AMR and/or identify factors associated with them. The study protocol was developed and registered in an international database, the Prospective Register of Systematic Reviews, by the University of York Centre for Reviews and Dissemination (available at: *https://www.crd.york.ac.uk/prospero/ #recordDetails*, identifier: CRD42022380207) to promote and guarantee transparency in the systematic review process, minimize the risk of reporting bias and protect from duplication problems. Besides, a protocol amendment was made regarding the title, and records were submitted online to the CRD editorial team. To guide the writing and registration procedures of the protocol, a seventeen-item Preferred Reporting Items for Systematic Reviews and Meta-Analyses Protocols 2015 checklist was used [20]. Finally, the results of the review were reported using the Preferred Reporting Items for Systematic Reviews and Meta-Analyses (PRISMA) 2020 checklist (**S1 Checklist**).

### Inclusion and exclusion criteria

The methodological guidance for systematic reviews of observational epidemiological studies reporting prevalence and cumulative incidence data was followed for determining the eligibility criteria, which were based on the CoCoPop mnemonic (condition, context, and population) [21]. **Population/Participants**: We included primary studies involving various participants (patients, the general population, healthcare workers, livestock producers, and tertiary-level students). **Condition**: This systematic review and meta-analysis considered studies that had measured the outcome of interest based on knowledge, attitudes, and practices toward AMR and /or factors associated with them. **Context/settings**: All observational epidemiological studies (case-control, cohort, and cross-sectional) conducted in Ethiopia and published in English between January 2010 and December 2022 in international and domestic peer-reviewed journals were included. Furthermore, studies without full-text access; articles that contained insufficient information on the outcomes of the interest (knowledge, attitudes, and practices towards AMR); studies not available as free full-text; findings from personal opinions; articles reported outside the scope of the outcome of interest; qualitative study design; case reports; case series; letters to editors; and unpublished data were excluded.

### Information sources and search strategy

Literature search strategies were developed using medical subject headings (MeSH) and text words related to the outcomes of the study. The search typically included the following electronic bibliographic databases: Excerpta Medica database, PubMed, Web of Science, African Journal of Online, Google Scholar, Google, and Cochrane Library to ensure complete coverage of the topic by accounting for variability between the indexing in each database. The literature search was limited to studies published in the English language between January 2010 and December 2022 which explored Ethiopians' knowledge, attitudes, and practices toward AMR and/or factors associated with them. The reference lists of included studies identified through the search were scanned to ensure literature saturation. Where necessary, we also searched the

authors' files to ensure that all relevant materials had been captured. For the advanced search in PubMed, the following steps comprised the search process: Initially, the search terms were developed along five domains: "Antimicrobial resistance", "knowledge", "attitude"," practices"," and associated factors". As such, we gathered keywords from Google Scholar, Wikipedia, and Google for each concept, which was then searched independently in PubMed to find MeSH terms in the MeSH hierarchy tree and then combined in an advanced search. Boolean logic ("AND" and "OR") was used to combine these concepts. The search was double-blinded and conducted from 1st, October to 31st, December 2022, by two authors (BW and MO). The PubMed search terms with their Boolean operators of this study were supplied as an additional file (S1 Table).

## Study selection procedures

The articles that were found through the electronic database searches were exported to the reference management software, EndNote X7, where duplicate studies were then eliminated. Two authors (BW and MO) independently screened the titles and abstracts that were obtained by the search against the inclusion criteria. To describe the extent to which assessments by multiple authors are similar, inter-rater agreement was calculated after referring to the Cochrane Handbook for systematic reviews [22]. Thus, a value of kappa 0.75 or more was considered, reflecting excellent agreement. The screened articles were then subjected to a full article review by two independent authors (TM and AK). Pre-specified criteria for inclusion in the review were followed to determine which records were relevant and should be included. Where more information was required to answer queries regarding eligibility, the remaining authors were involved. Disagreements about whether a study should be included were resolved by discussion. Moreover, the reasons for excluding the articles were recorded at each step.

## Data extraction

Two authors (BW and MO) working independently abstracted the relevant data from studies by using a standardized Microsoft Excel spreadsheet. For data extraction, Joana Briggs Institute (JBI) adopted data collection formats suitable for meta-analysis were employed [23]. The first author's last name, sample characteristics, year of publication, study design, study setting, timing and procedures of data collection, response rates, and outcome measures were collected. The reliability agreement among the data extractors was evaluated and verified using Cohan's kappa coefficient after data was recovered from 30% of the primary studies [24].

As a consequence, the kappa coefficient's strength of agreement was divided into four categories: low (0.20), fair (0.21–0.40), moderate (0.41–0.60), good (0.61–0.80), and virtually perfect agreement (0.81–1). A kappa statistic value of more than or equal to 0.5 was regarded as congruent and acceptable. In the case of disagreements between the two data extractors, a third author (TM) was involved in adjudicating unresolved disagreements through discussion and re-checking of the original articles.

## Outcome measurement

The outcome measurements of this review were reported in terms of the percentages of participants with a good level of knowledge, favorable attitudes, and good practices toward AMR. Furthermore, associated factors were narrated in texts, as we identified insufficient data on factors influencing the outcome to pool the odds ratio (OR), and the included primary studies that assessed the associated factors had heterogeneous explanatory variable classifications concerning the outcome variables.

## Methodological quality (risk of bias) assessment

To assess the quality of the studies, the JBI critical appraisal checklists for cross-sectional study (analytical or descriptive) were employed [25]. Two authors (BW, and MO) independently assessed the methodological quality of each study. In this manner, the following major components were evaluated for studies reporting descriptive cross-sectional data only [26, 27]: Q1) Was the sample frame appropriate to address the target population? Q2) Were study participants sampled in an appropriate way? Q3) Was the sample size adequate? Q4) Were the study subjects and the setting described in detail? Q5) Was the data analysis conducted with sufficient coverage of the identified sample? Q6) Were valid methods used for the identification of the condition? Q7) Was the condition measured in a standard, reliable way for all participants? Q8) Was there appropriate statistical analysis? Q9) Was the response rate adequate, and if not, was the low response rate managed appropriately? (**S2A Table**). In addition, the JBI checklist assessed the following major components for the analytical cross-sectional studies [28–39]: Q1) Were the criteria for inclusion in the sample clearly defined? Q2) Were the study subjects and the setting described in detail? Q3) Was the exposure measured in a valid and reliable way? Q4) Were objective, standard criteria used for the measurement of the condition? Q5) Were confounding factors identified? Q6) Were strategies to deal with confounding factors stated? Q7) Were the outcomes measured in a valid and reliable way? Q8) Was appropriate statistical analysis used? (**S2B Table**). Response options to corresponding questions in both critical appraisal tools were "yes," "no," "unclear," and "not applicable." An Overall score was calculated by counting the number of "yes"s in each row. For the final decision, the number of "yes" responses was counted for individual articles. After being evaluated against these criteria, articles with an overall appraisal score of 7 or higher were included in the quantitative analysis. Disagreements were resolved by consulting with a third independent author (AK) when this occurred.

## Data synthesis and meta-analysis

The extracted data were imported from a Microsoft Excel spreadsheet into STATA MP 16 statistical software (StataCorp LP, 4905 Lakeway Drive, College Station, TX 7845, USA) for analysis. The heterogeneity of the results was visually examined via the forest plots with pooled estimates. Thus, its presence was confirmed subjectively with a lack of overlap between the confidence interval (CI). In addition, the statistical heterogeneity was explored more formally by using Cochran's Q test ($x^2$) at P-value < 0.10 indicating significant heterogeneity, and Higgins and Thompson's $I^2$ statistics, were employed to estimate the percentages of the between-study variability where,0%,25 to 50%,50 to 75%, and greater than or equal to75% indicated no heterogeneity, low heterogeneity, moderate heterogeneity, and high heterogeneity respectively [40]. The random-effect meta-analysis model was used to estimate Der Simonian and Laird's pooled effect due to the presence of considerable statistical heterogeneity [41]. Subgroup meta-analysis based on the participants as covariates, meta-regression, and sensitivity analyses were also performed to investigate the source of statistical heterogeneity. Publication or dissemination bias was examined subjectively using funnel plots and objectively using the nonparametric rank correlation test of Begg [42] with *P* <0.05 being taken into consideration to declare potential publication bias. Results were presented in the form of tables, texts, and figures.

# Results

## Search and study selection

Our search was restricted to articles published in the English language between January 2010 and December 2022 in the electronic databases of PubMed, Web of Science, and Excerpta

Medica database. In addition, Google, Google Scholar, and African Journal of Online were searched. Through systematic and manual searching, 700 primary articles were found. Due to duplication, 610 articles were removed. The remaining 90 were screened based on their title and abstract, with 60 being eliminated as unrelated to our study. Finally, 30 full-text primary articles were evaluated against eligibility criteria, and 14 were selected for quantitative analysis (**Fig 1**).

## Study characteristics

A total of 14 published studies involving 4476 study subjects were included in this systematic review and meta-analysis. The sample size of the primary studies included in the review was significantly variable and ranged from (n = 91) [33] to (n = 571) [27]. Among the primary studies that reported gender [26–38], the majority of study participants (n = 2409) were male. All studies employed cross-sectional research designs [26–39]. The authors of the primary studies recruited study participants from healthcare workers [26, 29, 37–39], patients [28, 32], tertiary school students [30, 35], community members /the general public [31, 36], and live-stock owners [27, 33, 34]. The sampling techniques involved probability sampling techniques [26, 27, 31–33, 35–37, 39] and non-probability sampling techniques [28–30, 34, 38] to choose a representative sample. The majority of the studies included in this review were research reports from North Ethiopia's Amhara Region State [26–28, 30–34, 36, 38, 39]. Fifty percent of the survey tool was a self-administered, standardized questionnaire [26, 29, 30, 33, 35, 38, 39]. The percentages of a good level of knowledge, favorable attitudes, and good practices toward AMR range from 18.47% [34] to 84.71% [26], 14.71% [27] to 96.28% [30], and 37.30 [28] to 65.58% [38] respectively. Except for a research report by Bayeh et al. [39], which was published in 2014, all were published articles in the past six years (2017–2022) [26–38] (**Table 1**).

## Knowledge, attitudes and practices toward antimicrobial resistance

A meta-analysis was conducted on thirteen studies that reported AMR knowledge [26–36, 38, 39], ten studies that assessed attitudes toward AMR [27, 29–34, 36–38], and five studies that reported AMR practices [28, 32, 33, 36, 38]. Therefore, given the substantial statistical heterogeneity in the fixed-effects model, the pooled estimate was determined using a random-effects model. We found that the overall pooled level of good knowledge regarding AMR was only 51.53% [(95% CI: 37.85, 65.21), $I^2$ = 99.0%, P <0.001]. Furthermore, an overall pooled level of favorable attitudes and AMR practices were 63.43% [(95% CI: 42.66, 84.20), $I^2$ = 99.6, P <0.001] and 48.85% [(95% CI: 38.68, 59.01), $I^2$ = 93.1, P <0.001], respectively (**Fig 2A** to **2C**).

## Subgroup meta-analysis

The results of the subgroup statistical analysis of this meta-analysis (stratified by population) revealed that the level of good knowledge of AMR among healthcare workers was 74.21% (95% CI: 65.44, 82.98), the Higgins and Thompson's $I^2$ statistics showed high heterogeneity ($I^2$: 92.9%),P<0.001), and 56.46% [95% CI: 33.71, 79.21), the Higgins and Thompson's $I^2$ statistics showed high heterogeneity ($I^2$ = 96.8%), P< 0.001) among tertiary school students. Similarly, the highest level of favorable attitudes was observed among healthcare workers, 83.89 (95% CI: 70.29, 97.49), the Higgins and Thompson's $I^2$ statistics showed high heterogeneity ($I^2$ = 96.7%), P< 0.001). In contrast, good AMR practices were low among all participants (**Table 2**).

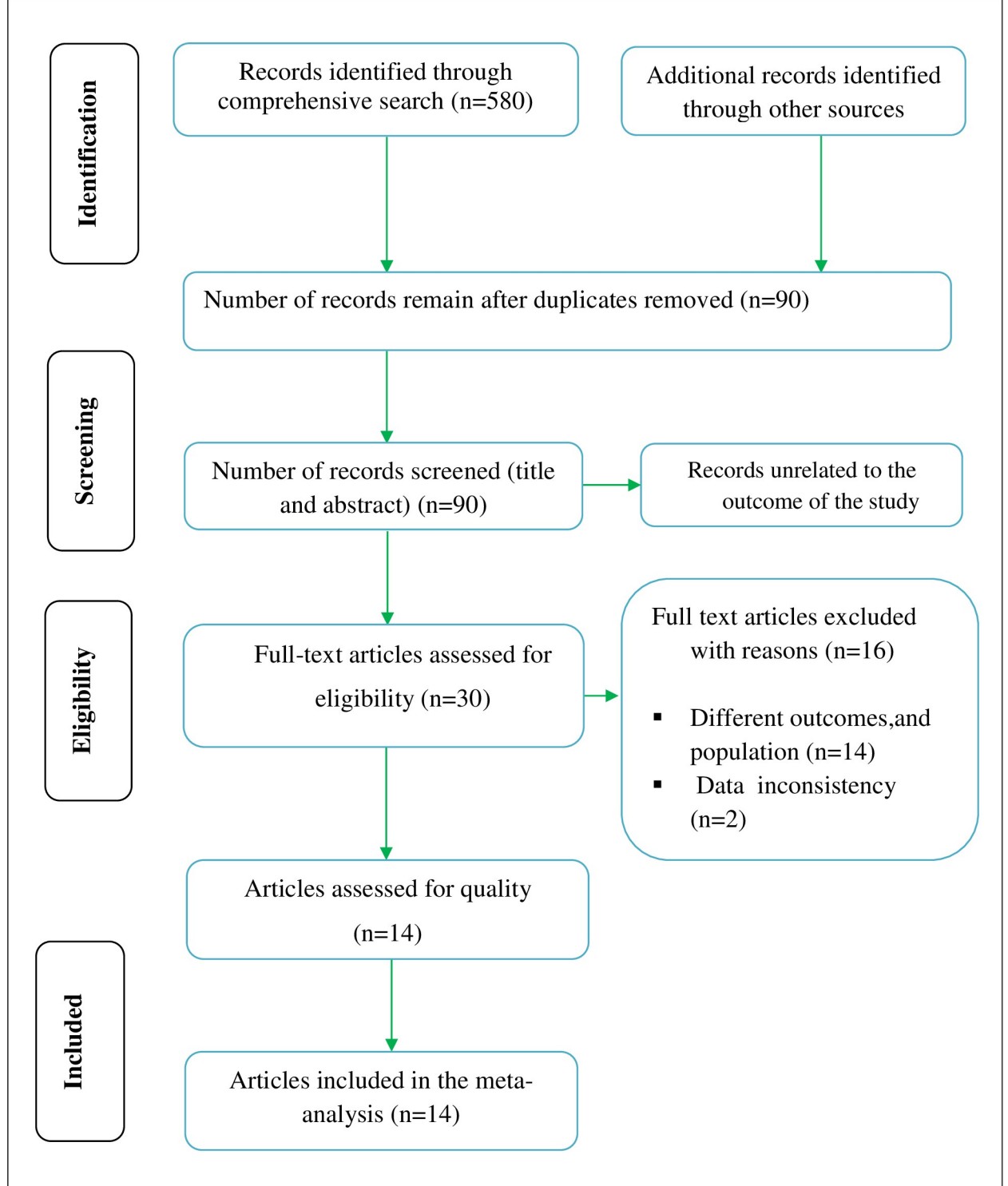

**Fig 1. PRISMA flow diagram explaining the selection of primary studies.**

**Table 1. The characteristics of the studies included in the systematic review and meta-analysis.**

| S. No | Author | Year | Region | Study design | Sample size | Good knowledge, n (%) | Favorable attitudes, n (%) | Good practices, n (%) | Response rate | Appraisal score |
|---|---|---|---|---|---|---|---|---|---|---|
| 1 | Bayeh et al. [39] | 2014 | Amhara | Cross-sectional | 385 | 278 (72.21) | NR | NR | 91.4% | 9 |
| 2 | Belachew et al. [38] | 2022 | Amhara | Cross-sectional | 276 | 210 (76.09) | 198 (71.74) | 181 (65.58) | 86.6% | 9 |
| 3 | Beyene et al. [37] | 2017 | Addis Ababa | Cross-sectional | 205 | NR | 195 (95.12) | NR | NR | 9 |
| 4 | Dejene et al. [36] | 2022 | Amhara | Cross-sectional | 400 | 180 (45.00) | 215 (53.75) | 200 (50.00) | NR | 9 |
| 5 | Fetensa et al. [35] | 2020 | Amhara | Cross-sectional | 232 | 158 (64.10) | NR | NR | 93.6% | 9 |
| 6 | Gebeyehu et al. [27] | 2021 | Amhara | Cross-sectional | 571 | 110 (19.26) | 84 (14.71) | NR | 100% | 8 |
| 7 | Gemeda et al. [34] | 2020 | Amhara and Oromia | Cross-sectional | 379 | 70 (18.47) | 263 (69.39) | NR | NR | 9 |
| 8 | Geta and Kibret [33] | 2021 | Amhara | Cross-sectional | 91 | 46 (50.55) | 48 (52.75) | 43 (47.25) | 100% | 9 |
| 9 | Geta and Kibret [32] | 2022 | Amhara | Cross-sectional | 232 | 87 (37.50) | 105 (45.26) | 102 (43.97) | NR | 9 |
| 10 | Mengesha et al. [31] | 2020 | Amhara | Cross-sectional | 374 | 160 (42.78) | 194 (50.52) | NR | 97.0% | 8 |
| 11 | Seid et al. [30] | 2018 | Amhara | Cross-sectional | 323 | 145 (44.89) | 311 (96.28) | NR | 90.0% | 9 |
| 12 | Simegn et al. [26] | 2022 | Amhara | Cross-sectional | 412 | 349 (84.71) | NR | NR | 97.4% | 8 |
| 13 | Tafa et al. [29] | 2017 | Dire Dawa | Cross-sectional | 218 | 137 (62.84) | 184 (84.40) | NR | NR | 8 |
| 14 | Tesfaye et al. [28] | 2017 | Amhara | Cross-sectional | 378 | 180 (47.62) | NR | 141 (37.30) | 91.8% | 9 |

NR,not reported; %,percentage; n, numbers of cases

## Meta-regression

Due to limited sample sizes and the number of studies included in this systematic review and meta-analysis, we could not complete meta-regression analyses regarding practices towards AMR, however, random-effects meta-regression using sample size and year of publication as covariates were performed to explore the source of heterogeneity at a 5% significance level for a good level of knowledge and favorable attitudes towards AMR. As shown in **Table 3,** these covariates were not found to be the source of heterogeneity.

## Sensitivity meta-analyses

A leave-out-one sensitivity analysis was conducted to assess the impact of each study on the pooled level of good knowledge about, favorable attitudes and good practices toward AMR while gradually excluding each study. Results showed that the combined effects did not significantly change as a result of the excluded study (**Table 4**).

## Publication bias (Reporting bias)

Publication bias was assessed using funnel plots and by the regression-based test of Egger at $P<0.05$. On visual inspection, the funnel plot showed slight asymmetrical distribution (**Fig 3A**

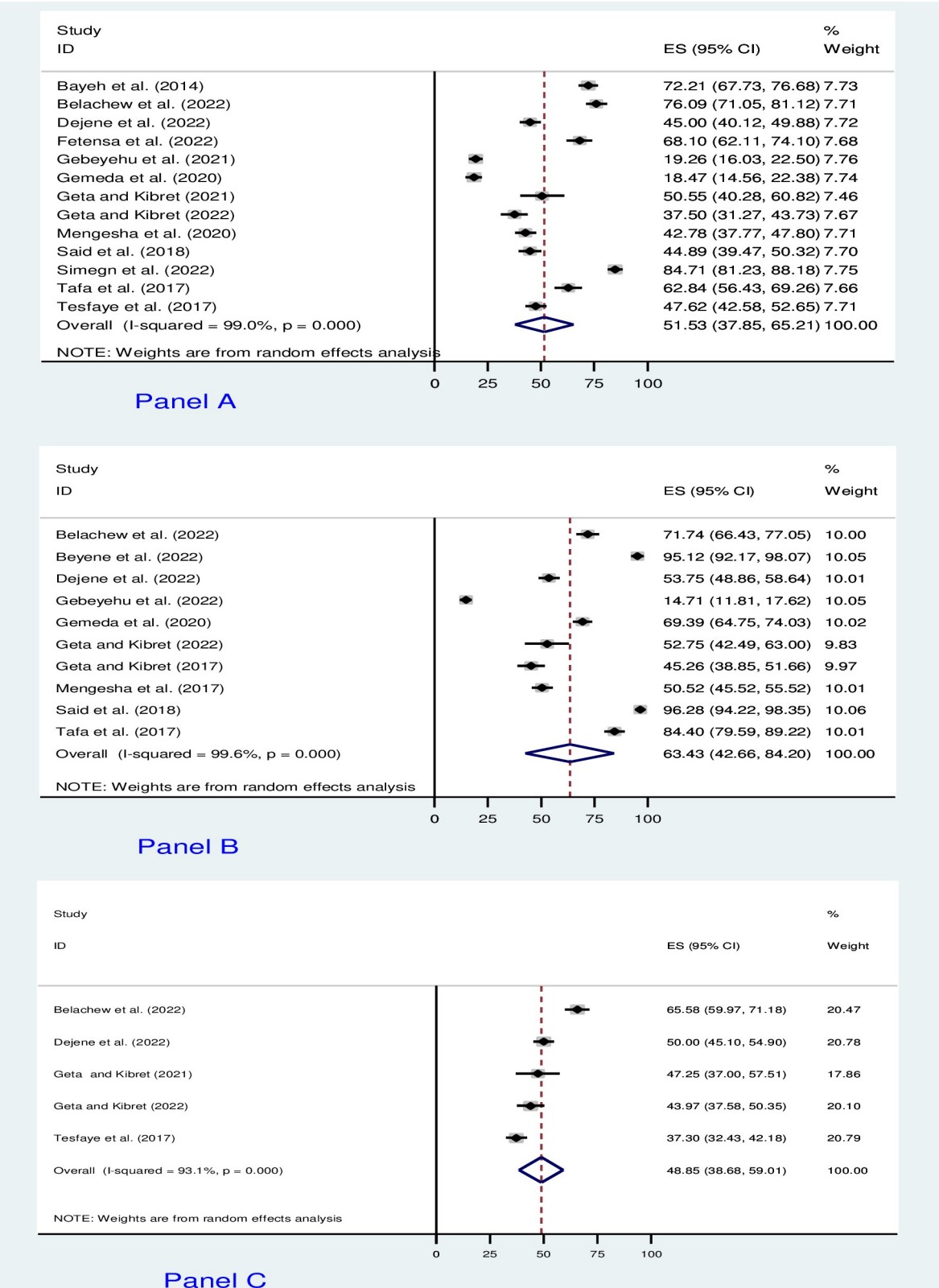

**Fig 2. The forest plot of pooled prevalence.** (Panel A) Good level of AMR knowledge. (Panel B) Favorable attitudes towards AMR. (Panel C) Good AMR practices.

**Table 2. Results of subgroup meta-analysis of good knowledge, favorable attitudes, and good practices.**

| Outcome | Participant's characteristics | Number of study | Total Participants | Effect size (95% CI) | Heterogeneity | |
|---|---|---|---|---|---|---|
| | | | | | $I^2$ value | *P*.value |
| Good knowledge | Healthcare workers [26, 29, 38, 39] | 4 | 1291 | 74.21(65.44, 82.98) | 92.9% | <0.001 |
| | Tertiary school students [30, 35] | 2 | 555 | 56.46 (33.71, 79.21) | 96.8% | <0.001 |
| | General population [31, 36] | 2 | 774 | 43.92(40.43, 47.42) | 00.0% | 0.534 |
| | Patients [28, 32] | 2 | 610 | 42.73(32.82, 52.64) | 83.7% | 0.013 |
| | Livestock producers [27, 33, 34] | 3 | 1041 | 27.91(16.44, 39.38) | 96.8% | <0.001 |
| Favorable attitudes | Healthcare workers [29, 37, 38] | 3 | 699 | 83.89 (70.29, 97.49) | 96.7% | <0.001 |
| | General population [31, 36] | 2 | 784 | 52.17 (48.68, 55.67) | 00.0% | 0.365 |
| | Livestock producers [27, 33, 34] | 3 | 1041 | 45.55(4.86,86.23) | 99.5% | <0.001 |
| | Others (patients and tertiary school students [30, 32] | 2 | 555 | 70.87 (20.86,120.87) | 99.5% | <0.001 |
| Good practices | Patients [28, 32] | 2 | 610 | 40.30 (33.80, 46.80) | 62.2% | 0.104 |
| | Others (general population, healthcare workers, and livestock producers) [33, 36, 38] | 3 | 767 | 54.64 (42.89, 66.38) | 89.9% | <0.001 |

CI,confidence interval; Higgins and Thompson's $I^2$ statistics estimated the percentages of the between-study variability: no heterogeneity (0%),low heterogeneity(25 to 50%), moderate heterogeneity (50 to 75%), and high heterogeneity (greater than or equal to75%); P.value < 0.10 indicates significant heterogeneity,formal way of exploring statistical heterogeneity using Cochran's Q test ($x^2$).

to **3C**). However, the formal Egger linear regression test was not statistically significant for a good level of AMR knowledge (t = 0.58, P = 0.571), favorable AMR attitude (t = -0.98, P = 0.357), and good AMR practices (t = 0.14, P = 0.900) corroborating the absence of evidence of small study effects.

### Factors associated with knowledge, attitudes, and practices toward AMR

Although evidence is limited, two studies [26, 27] found that being male, and having attended formal education were positively associated with KAP towards AMR. Furthermore, a research report by Simegn et al. revealed that having experienced antibiotic use was associated with a good level of knowledge about AMR [26] (**Table 5**).

## Discussion

This systematic review and meta-analysis is the first of its kind and presents the most comprehensive assessment of the level of good knowledge, favorable attitudes, and good practices toward AMR. We included 14 eligible studies with 4476 participants involving healthcare

**Table 3. Meta-regression analysis of factors affecting between-study heterogeneity.**

| Heterogeneity source | Coefficient | Standard Error | t | P> t | 95% Confidence interval | |
|---|---|---|---|---|---|---|
| **Level of good AMR knowledge** | | | | | | |
| Sample size | -.056044 | .0518812 | -1. 08 | 0. 305 | –. 1716426 | .0595545 |
| Year of publication | -.9507475 | 2.390027 | -0.40 | 0. 699 | –6.276061 | 4.374565 |
| **Level of favorable attitude toward AMR** | | | | | | |
| Sample size | -.0957215 | .0607642 | -1.58 | 0.159 | -.2394061 | .0479631 |
| Year of publication | -1.358004 | 3.412778 | -0.40 | 0.703 | -9.427942 | 6.711935 |

**Table 4. Sensitivity analysis of pooled prevalence with each study removed one by one.**

| Study omitted | Estimate | 95% Confidence interval | |
|---|---|---|---|
| **Good level of knowledge about AMR** | | | |
| Bayeh et al. (2014) [39] | 49.79821 | 35.465099 | 64.131325 |
| Belachew et al. (2022) [38] | 49.477707 | 35.333866 | 63.621544 |
| Dejene et al. (2022) [36] | 52.076485 | 37.226902 | 66.926071 |
| Fetensa et al. (2022) [35] | 50.151756 | 35.719959 | 64.583557 |
| Gebeyehu et al. (2021) [27] | 54.246498 | 41.368633 | 67.124359 |
| Gemeda et al. (2020) [34] | 54.305344 | 40.847694 | 67.762993 |
| Geta and Kibret (2021) [33] | 51.609097 | 37.240738 | 65.977455 |
| Geta and Kibret (2022) [32] | 52.695576 | 38.156399 | 67.234749 |
| Mengesha et al. (2020) [31] | 52.261505 | 37.46896 | 67.054054 |
| Said et al. (2018) [30] | 52.084095 | 37.351143 | 66.817039 |
| Simegn et al. (2022) [26] | 48.729164 | 36.403912 | 61.054417 |
| Tafa et al. (2017) [29] | 50.591312 | 36.086861 | 65.095764 |
| Tesfaye et al. (2017) [28] | 51.857327 | 37.028885 | 66.685768 |
| Combined | 51.529733 | 37.84905 | 65.210417 |
| **Favorable attitude toward AMR** | | | |
| Belachew et al. (2022) [38] | 62.502693 | 39.809155 | 85.196228 |
| Beyene et al. (2022) [37] | 59.887157 | 36.881901 | 82.892418 |
| Dejene et al. (2022) [36] | 64.503197 | 41.903093 | 87.103004 |
| Gebeyehu et al. (2022) [27] | 68.981995 | 55.534237 | 82.429756 |
| Gemeda et al. (2020) [34] | 62.761555 | 39.81673 | 85.706383 |
| Geta and Kibret (2022) [33] | 64.592613 | 42.564278 | 86.620949 |
| Geta and Kibret (2017) [32] | 65.440742 | 43.300583 | 87.580902 |
| Mengesha et al. (2017) [31] | 64.862457 | 42.387482 | 87.337433 |
| Said et al. (2018) [30] | 59.75563 | 38.22723 | 81.284027 |
| Tafa et al. (2017) [29] | 61.092407 | 38.359745 | 83.825073 |
| Combined | 63.428928 | 42.662573 | 84.195283 |
| **Good level of AMR practices** | | | |
| Belachew et al. (2022) [38] | 44.386726 | 37.87331 | 50.900143 |
| Dejene et al. (2022) [36] | 48.538776 | 34.635681 | 62.441872 |
| Geta and Kibret (2021) [33] | 49.194901 | 37.327873 | 61.061928 |
| Geta and Kibret (2022) [32] | 50.067833 | 37.379513 | 62.756153 |
| Tesfaye et al. (2017) [28] | 51.925362 | 41.908466 | 61.942257 |
| Combined | 48.84628 | 38.682999 | 59.009562 |

workers, the general population, tertiary-level health sciences and medical students, and livestock and producers in various settings of Ethiopia.

To begin with, researchers have described that irrational antimicrobial prescribing and dispensing practices in low- and middle-income countries, Ethiopia included, are frequently attributed to a lack of knowledge about AMR [43, 44]. Of the 14 studies included in this systematic review and meta-analysis, knowledge about AMR was assessed in 92.86% (13 out of 14) of them with 4271 participants. As a result, approximately only half (52%) of Ethiopians had a good level of knowledge about AMR. When participants' characteristics are stratified, the general population (43.92%), patients visiting healthcare facilities (42.73%), and livestock producers (27.11%) had a lower level of good knowledge of AMR.

About the general population, this review's finding was lower than a research report from Japan, where approximately half of the public had adequate knowledge and awareness about

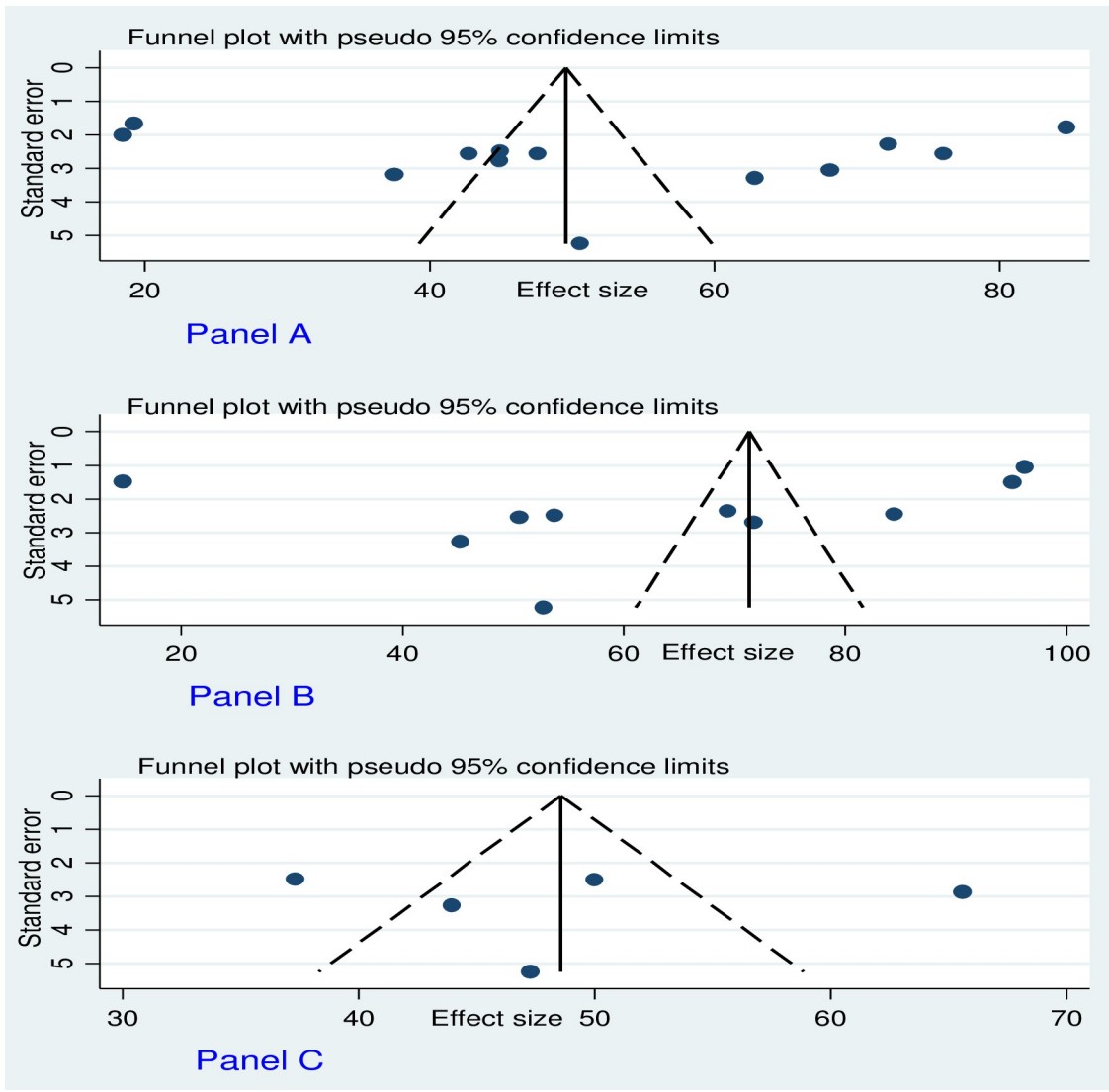

**Fig 3. Funnel plots of publication bias.** (Panel A) For a good level of AMR knowledge. (Panel B) Favorable attitudes towards AMR. (Panel C) A good level of AMR practices. The Y-axis shows the standard error of the estimate (prevalence). The 95% confidence interval is represented by the dashed lines. The X-axis represents the estimate (i.e. prevalence) and the dots show the distribution of individual studies.

AMR [17]; the Western region of Saudi Arabia (55%) [45, 46]; the general rural population in low-income and middle-income in South-East Asia: Chiang Rai, Thailand (74.8%), and Salvan (62.5%) [47]. Furthermore, in a meta-analysis of 24 studies comprising 40747 general population, Gualano et al., found that approximately 59% of people were aware of AMR [46]. Nevertheless, the pooled random effect mata-analysis of our study was higher than reports in Nigeria, where only 8.3% of the population was reported to have exhibited good knowledge of AMR [48]. These disparities can be explained by the size of study participants, access to social media and traditional media channels, the country's commitment to the implementation of the antibiotic stewardship program, and other related policies on antibiotic use and resistance. In addition, many antimicrobials are used by farmers, particularly livestock producers [49]. In keeping with this, the assessment of livestock producers' KAP, on the other hand, is the

**Table 5. Factors associated with KAP towards AMR in studies using regression models.**

| Authors | Outcome | Regression model fitted | key findings |
|---|---|---|---|
| Gebeyehu et al [27] | Knowledge, Attitudes, and Practices | Binary logistic regression | The educational level of animal producers was negatively correlated (adjusted odds ratio (AOR<1) with their KAP towards AMR; only the variable "sex" was positively correlated with both the knowledge (AOR: 3.49) and the practice (AOR: 2.89) among animal producers. This implies that male animal producers were more knowledgeable about AMR than female producers. |
| Simegn et al [26] | Knowledge | Binary logistic regression | Being male (AOR:1.94)95% CI: 1.10, 3.52);having 6–10 years of work experience (AOR:2.45 (95% CI: 1.28, 4.68);30–38 working hours per week (AOR:3.93 (95% CI: 1.38, 5.11.12),and history of antibiotics intake (AOR: 3.71(95% CI: 1.75, 7.87) were significantly associated with good Knowledge towards AMR. |

Abbreviations: CI, confidence interval; AOR, adjusted odds ratio; KAP, knowledge, attitudes, and practices; AMR, antimicrobial resistance

foundation for raising awareness and developing appropriate legislation regarding antimicrobial use and resistance as Ethiopia has the largest livestock production in Africa [27]. Albeit, only approximately 28% of the livestock producers had a good level of AMR knowledge according to our study. In contrast, in this review, about 74% of healthcare workers had a good level of AMR knowledge, which is by far higher than the findings of a national survey in Nigeria, where only 49.2% had adequate knowledge about AMR, with physicians having significantly better knowledge than other health care workers [50], and almost in line with a study that assessed the KAP of health workers in 30 European Union/European Economic Area countries concerning AMR [51]. A higher pooled prevalence of good knowledge of AMR observed in this study and reports from other countries [52, 53] can be justified by the fact that educational training programs have been shown to improve healthcare professionals' knowledge of antimicrobial stewardship [54].

Concerning participants' attitudes, this systematic review and meta-analysis found that more than 63% of participants had favorable attitudes toward AMR. This finding is consistent with previous research findings [55, 56]. This could be due to a higher proportion of healthcare workers knowing about AMR compared to the general population (44%), patients (43%), and livestock keepers (28%) resulting in favorable attitudes.

Moreover, we computed the pooled estimate from data extracted from five studies involving 1377 participants and determined that the combined proportion of Ethiopians with good AMR practices was lower than average (49%) with an even lower estimate among patients (40%). Dispensing antibiotics without a prescription on a direct request from a client [38], purchasing without obtaining a doctors consultation [36], treating animals with antimicrobials prescribed for humans [36], and failing to take a full course of antibiotics [32, 33] were some of the prominent poor practices reported. Such poor or low levels of good AMR practices were consistent with a review report by Belachew et al. in Sub-Saharan Africa where non-prescribed dispensing of antibiotics was found to be a common practice among community drug outlets [57]. The observed poor AMR practices can be reflected by a significant knowledge gap about AMR reported earlier in this review. Of the 14 studies included in this systematic review and meta-analysis, factors associated with KAP were reported in 2 studies [26, 27]. According to Gebeyehu et al., animal producers with lower academic levels were unaware of AMR [27]. Pham-duc et al. reported similar findings in which Vietnamese with formal education were knowledgeable about AMR [58]. This could be due to individuals having access to read and get information, as well as explore other sources of information with ease. Furthermore, male participants had positive and statistically significant associations with KAP concerning AMR [26, 27]. Similar study findings were reported in China [59]. The reason could be that men are more likely to be exposed to public meetings and media and spend much of the day outdoors.

## The strength and limitations of this study

This study avoided duplication of similar work because the protocol for it was registered. The current review is novel as it revealed current evidence regarding KAP of AMR from both human and animal health, and therefore it will influence future public health policy toward working with animal and public health experts as AMR serious global and national health threat in human and livestock production. A double-blinded comprehensive search was conducted over a reputable period in more than seven online databases to avoid missing published studies. In addition, more than two data abstractors were involved, and to ensure inter-rater agreement, we consulted the Cochrane Handbook for systematic reviews. The newly amended JBI critical appraisal tool was used for quality assessment. Further analyses were conducted to explore sources of dissemination or publication biases. We followed the updated 2020 PRISMA checklist to compile the report. The limitations of this systematic review have also been acknowledged. The majority of the studies included in the current meta-analysis were conducted in the Amhara region. It is evident that it may lack national representativeness because little information was found in the remaining regions of Ethiopia, however, there may be no such socio-cultural differences across regions. Furthermore, the results of this review should be interpreted with caution due to significant heterogeneity in pooled effect estimates. The determinant factors meta-analysis was not pooled due to limited studies that investigated factors associated with KAP of AMR.

## Conclusion and recommendations

Responding to AMR threats requires policymakers, researchers, intersectoral and multidisciplinary collaboration, and local and international stakeholders to assess the population's KAP toward AMR. The pooled random effect meta-analysis in this review revealed a significant knowledge and practice gap regarding AMR among the general public, patients, and livestock producers. As a consequence, targeted educational interventions need to be in place to raise individuals' understanding of AMR and to develop effective AMR countermeasures. Furthermore,the current review provides an opportunity to conduct a national wide survey to address the limitations identified in the review.

## Supporting information

**S1 Checklist. PRISMA 2020 checklist.**
(DOCX)

**S1 Table. Search details.**
(DOCX)

**S2 Table. JBI's critical appraisal tools: (A) Descriptive cross-sectional studies.** (B) Analytical cross-sectional studies.
(DOCX)

**S1 File. Data extraction sheet.**
(XLSX)

## Acknowledgments

We would like to thank the authors of the original studies included in this systematic review and meta-analysis.

## Author Contributions

**Conceptualization:** Beshada Zerfu Woldegeorgis, Mohammed Suleiman Obsa, Taklu Marama Mokonnon.

**Data curation:** Beshada Zerfu Woldegeorgis, Amene Abebe Kerbo, Mohammed Suleiman Obsa.

**Formal analysis:** Beshada Zerfu Woldegeorgis, Amene Abebe Kerbo, Mohammed Suleiman Obsa, Taklu Marama Mokonnon.

**Methodology:** Beshada Zerfu Woldegeorgis, Taklu Marama Mokonnon.

**Project administration:** Beshada Zerfu Woldegeorgis.

**Resources:** Beshada Zerfu Woldegeorgis.

**Software:** Beshada Zerfu Woldegeorgis.

**Supervision:** Beshada Zerfu Woldegeorgis.

**Visualization:** Beshada Zerfu Woldegeorgis, Taklu Marama Mokonnon.

**Writing – original draft:** Beshada Zerfu Woldegeorgis.

**Writing – review & editing:** Beshada Zerfu Woldegeorgis, Amene Abebe Kerbo, Mohammed Suleiman Obsa, Taklu Marama Mokonnon.

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
