## [Decision Letter · Decision Letter 0]

19 Apr 2023

PONE-D-23-06485A systematic review and meta-analysis of antimicrobial resistance knowledge, attitudes, and practices: current evidence to build a strong national antimicrobial resistance narrative in EthiopiaPLOS ONE

Dear Dr. Woldegeorgis,

Thank you for submitting your manuscript to PLOS ONE. After careful consideration, we feel that it has merit but does not fully meet PLOS ONE’s publication criteria as it currently stands. Therefore, we invite you to submit a revised version of the manuscript that addresses the points raised during the review process.

Reviewer #1:

The authors reviewed cross-sectional studies in Ethiopia and extracted estimates from them to draw a conclusion in 3 aspects (knowledge, attitudes and practice) of AMR.

In general, the article is written clearly with straightforward objectives and step-by-step explanation of methodology.

The following points shall be addressed prior to publishing.

- citations should be placed on the first sentences.

e.g. [15] on page 5; [23] on page 10.

- on page 8, can the authors explain the reasons for excluding studies without full-text access?

Is it related to library access? Or the studies themselves?

- are the two kappa's on page 9 and 10 the same?

A citation is needed for the Cochrane handbook for systematic reviews and also the threshold at 0.75 on page 9.

The authors can consider showing the values of kappa’s from these two sections.

- on page 9, what is the 'pre-defined eligibility criterion'?

It is not clear how the 16 studies were excluded in this step.

- on page 10, are the associated factors corresponding to the last section of the results on page 20, i.e. sex and educational level?

- can the authors explain why the 14 studies split into two additional files (3 and 4)?

The additional file 3 is not finished.

Suggest to put the scores listed in the additional files on Table 1, rather than simply showing "low risk" in the last column.

- on page 12, citations of all those technological methods such as Cochran's Q test, Higgins and Thompson’s I2 statistics and Der Simonian and Laird’s pooled effect are needed.

- page 13, "Google" not mentioned on page 8.

- page 14, should it be n=517[32] instead of n=412[27]?

- page 16, should it be “favorable attitudes” instead of “favorable level” in the title of figure 3?

0 - 100% on the x axis needed in Figures 2, 3, and 4.

- page 16 & 17, suggest making a table for the section of “subgroup meta-analysis”.

- page 20, the authors can consider adding a caption of how to interpret/read the funnel plot in figures 5, 6, and 7.

e.g. 95% CIs are represented by the two dashed lines; x and y axis represent study result/log-odds-ratios and precision/standard error; what is "log(pr)" on the x axis?

The authors can consider to combine 3 figures into one. (And combine Figures 2, 3, 4 similarly).

- page 20, additional file 5: the citation numbers [25 & 30] should be [27 & 32] in the main text.

- page 27, ref #24: minor styling issue.

Reviewer #2:

Title: A systematic review and meta-analysis of antimicrobial resistance knowledge, attitudes, and practices: current evidence to build a strong national antimicrobial resistance narrative in Ethiopia

General comment

The title is very important and relevant to recommend for policy makers for increasing the public perceptions on prudent antibiotic use to reduce AMR. The authors should be incorporated the comments and improve the manuscript accordingly.

Abstract

It is good

Background

Good but it is better to rewrite the paragraphs consistently i.e., from general to specific or specific to general (world to Ethiopia or Ethiopia to the world). Example Page 5 paragraph 1 about UK, paragraph 2 about Ethiopian and the last paragraph about Europe, Pakistan and Japan.

The gaps, objective and significance of the review is well stated.

Methods

It is well organized

It is better to indicate time frame /period of review

Inclusion & exclusion criteria

Inclusion criteria/condition, knowledge about, delete the word about

Context/Setting: Justify your possible reasons why you include peer reviewed journals published between January 2010 and December 2022.

Exclusion of articles written in language other than English may have excluded relevant articles

Exclusion criteria: add studies not available as free full-text as exclusion criteria

Information sources and search strategy

Ethiopians’ knowledge of, attitudes delete the word of

Study selection procedures ok

Data extraction ok

Outcome measurement ok

Methodological quality (risk of bias) assessment ok

Data synthesis and Meta-Analysis ok

Result

Well organized

You have included studies in your review based on quality assessment. Results in table 1 indicated that the quality is low but quality assessment of studies using JBI’s critical appraisal tools showed the quality score is 9. It makes confusion, please make it clear.

The funnel plot and Egger test indicated different publication bias, how to reconcile these results.

Figure 3: Forest plot of pooled estimates for favorable level of AMR, please add attitude

Why you did not indicate the KAP difference among location/ region?

It is difficult to change the attitude of human on antibiotic use and AMR. In your finding the level of favorable attitude was higher than knowledge and practice. What shows this finding?

Discussion

Well organized

It is better to state the possible reasons/associated factors for significant knowledge and practice gap and favorable level of attitude regarding AMR in Ethiopia

The comparisons are quite superficial and there is little discussion of the reasons for similarity / difference to previous work. The discussion section would benefit greatly from a description of the limitations of the current work

The strength and limitations of this study well addressed

Conclusion and recommendations

your conclusion and recommendations are well organized.

Since almost all studies included in the review are done in Amhara region, your data did not enough to generalized /concluded and recommend the KAP of AMR in Ethiopia. So, it is better to rewrite this part accordingly.

Thank you very much!==============================

We look forward to receiving your revised manuscript.

Kind regards,

Mohammed Feyisso Shaka, MPH

Academic Editor

PLOS ONE

Journal Requirements:

https://www.frontiersin.org/articles/10.3389/fneur.2023.1086622/full

In your revision ensure you cite all your sources (including your own works), and quote or rephrase any duplicated text outside the methods section. Further consideration is dependent on these concerns being addressed.

4. We note that you have referenced (unpublished on page 8) which has currently not yet been accepted for publication. Please remove this from your References and amend this to state in the body of your manuscript: (ie “Bewick et al. [Unpublished]”) as detailed online in our guide for authors

6. We note that this manuscript is a systematic review or meta-analysis; our author guidelines therefore require that you use PRISMA guidance to help improve reporting quality of this type of study. Please upload copies of the completed PRISMA checklist as Supporting Information with a file name “PRISMA checklist”.

Reviewers' comments:

Reviewer's Responses to Questions

**Comments to the Author**

1. Is the manuscript technically sound, and do the data support the conclusions?

Reviewer #1: Yes

Reviewer #2: Yes

2. Has the statistical analysis been performed appropriately and rigorously?

Reviewer #1: Yes

Reviewer #2: Yes

3. Have the authors made all data underlying the findings in their manuscript fully available?

Reviewer #1: Yes

Reviewer #2: Yes

4. Is the manuscript presented in an intelligible fashion and written in standard English?

Reviewer #1: Yes

Reviewer #2: No

5. Review Comments to the Author

Reviewer #1: The authors reviewed cross-sectional studies in Ethiopia and extracted estimates from them to draw a conclusion in 3 aspects (knowledge, attitudes and practice) of AMR.

In general, the article is written clearly with straightforward objectives and step-by-step explanation of methodology.

The following points shall be addressed prior to publishing.

- citations should be placed on the first sentences.

e.g. [15] on page 5; [23] on page 10.

- on page 8, can the authors explain the reasons for excluding studies without full-text access?

Is it related to library access? Or the studies themselves?

- are the two kappa's on page 9 and 10 the same?

A citation is needed for the Cochrane handbook for systematic reviews and also the threshold at 0.75 on page 9.

The authors can consider showing the values of kappa’s from these two sections.

- on page 9, what is the 'pre-defined eligibility criterion'?

It is not clear how the 16 studies were excluded in this step.

- on page 10, are the associated factors corresponding to the last section of the results on page 20, i.e. sex and educational level?

- can the authors explain why the 14 studies split into two additional files (3 and 4)?

The additional file 3 is not finished.

Suggest to put the scores listed in the additional files on Table 1, rather than simply showing "low risk" in the last column.

- on page 12, citations of all those technological methods such as Cochran's Q test, Higgins and Thompson’s I2 statistics and Der Simonian and Laird’s pooled effect are needed.

- page 13, "Google" not mentioned on page 8.

- page 14, should it be n=517[32] instead of n=412[27]?

- page 16, should it be “favorable attitudes” instead of “favorable level” in the title of figure 3?

0 - 100% on the x axis needed in Figures 2, 3, and 4.

- page 16 & 17, suggest making a table for the section of “subgroup meta-analysis”.

- page 20, the authors can consider adding a caption of how to interpret/read the funnel plot in figures 5, 6, and 7.

e.g. 95% CIs are represented by the two dashed lines; x and y axis represent study result/log-odds-ratios and precision/standard error; what is "log(pr)" on the x axis?

The authors can consider to combine 3 figures into one. (And combine Figures 2, 3, 4 similarly).

- page 20, additional file 5: the citation numbers [25 & 30] should be [27 & 32] in the main text.

- page 27, ref #24: minor styling issue.

Reviewer #2: Title: A systematic review and meta-analysis of antimicrobial resistance knowledge, attitudes, and practices: current evidence to build a strong national antimicrobial resistance narrative in Ethiopia

General comment

The title is very important and relevant to recommend for policy makers for increasing the public perceptions on prudent antibiotic use to reduce AMR. The authors should be incorporated the comments and improve the manuscript accordingly.

Abstract

It is good

Background

Good but it is better to rewrite the paragraphs consistently i.e., from general to specific or specific to general (world to Ethiopia or Ethiopia to the world). Example Page 5 paragraph 1 about UK, paragraph 2 about Ethiopian and the last paragraph about Europe, Pakistan and Japan.

The gaps, objective and significance of the review is well stated.

Methods

It is well organized

It is better to indicate time frame /period of review

Inclusion & exclusion criteria

Inclusion criteria/condition, knowledge about, delete the word about

Context/Setting: Justify your possible reasons why you include peer reviewed journals published between January 2010 and December 2022.

Exclusion of articles written in language other than English may have excluded relevant articles

Exclusion criteria: add studies not available as free full-text as exclusion criteria

Information sources and search strategy

Ethiopians’ knowledge of, attitudes delete the word of

Study selection procedures ok

Data extraction ok

Outcome measurement ok

Methodological quality (risk of bias) assessment ok

Data synthesis and Meta-Analysis ok

Result

Well organized

You have included studies in your review based on quality assessment. Results in table 1 indicated that the quality is low but quality assessment of studies using JBI’s critical appraisal tools showed the quality score is 9. It makes confusion, please make it clear.

The funnel plot and Egger test indicated different publication bias, how to reconcile these results.

Figure 3: Forest plot of pooled estimates for favorable level of AMR, please add attitude

Why you did not indicate the KAP difference among location/ region?

It is difficult to change the attitude of human on antibiotic use and AMR. In your finding the level of favorable attitude was higher than knowledge and practice. What shows this finding?

Discussion

Well organized

It is better to state the possible reasons/associated factors for significant knowledge and practice gap and favorable level of attitude regarding AMR in Ethiopia

The comparisons are quite superficial and there is little discussion of the reasons for similarity / difference to previous work. The discussion section would benefit greatly from a description of the limitations of the current work

The strength and limitations of this study well addressed

Conclusion and recommendations

your conclusion and recommendations are well organized.

Since almost all studies included in the review are done in Amhara region, your data did not enough to generalized /concluded and recommend the KAP of AMR in Ethiopia. So, it is better to rewrite this part accordingly.

Thank you very much!

6. PLOS authors have the option to publish the peer review history of their article (what does this mean?). If published, this will include your full peer review and any attached files.

**Do you want your identity to be public for this peer review?** For information about this choice, including consent withdrawal, please see our Privacy Policy.

Reviewer #1: No

Reviewer #2: No

 **********

---

## [Author Response · Author response to Decision Letter 0]

29 Apr 2023

Point by point response letter to reviewer one

Greetings, dear Reviewer 1: we are writing to express our gratitude and appreciation for reviewing our manuscript, "A systematic review and meta-analysis of antimicrobial resistance knowledge, attitudes, and Practices: current evidence to build a strong national antimicrobial resistance narrative in Ethiopia." We thoroughly addressed specific points made and uploaded a marked-up copy (revised manuscript with tracked changes) as a "Revised Article with Changes Highlighted” file, and a clean copy as “manuscript” file. At this point, again, we would like to thank you for taking the time to provide invaluable comments to enrich the contents and make them more palatable to the scientific community in general. 

- citations should be placed in the first sentences. e.g. [15] on page 5; [23] on page 10.

• Response: reviewer’s suggestion is appropriate, we checked all of them across the manuscript and corrected them.Thank you

- on page 8, can the authors explain the reasons for excluding studies without full-text access? Is it related to library access? Or the studies themselves?

• Response: dear reviewer, after the titles were screened, two studies had no full texts to read and decide whether it fits our outcomes or not. It is not due to liabrary access. Thank you!

-(About measuring agreement)- Are the two kappa's on page 9 and 10 the same? A citation is needed for the Cochrane Handbook for systematic reviews and also the threshold at 0.75 on page 9. The authors can consider showing the values of kappa from these two sections.

• Response: Dear reviewer, the concept of using the kappa on page 9 and page 10 for different purposes . In the study selection section, we used values of computed Kappa (such as between 0.40 and 0.59 to reflect the fair agreement, between 0.60 and 0.74 to reflect the good agreement and 0.75 or more to reflect excellent agreement) to measure the agreements between two authors (BW and MO) during the process of deciding which studies should be included in the study. Nevertheless, the second value of Kappa (page 10) was used to measure the level of agreement between the two authors (BW and MO) during data collection which involves extraction of specific data (estimates and others ) for meta-analysis based on the outcome of our study. Moreover, the decision values of kappa are alike. Finally, the sources referred to have been cited for both of them. Thank you 

- on page 9, what is the 'pre-defined eligibility criterion'?

• Response: At this step we mean the pre-specified criteria (based on condition, context, and population,) for inclusion in the review were followed. The phrase “A pre-defined eligibility criterion” creates ambiguity so it was deleted and replaced with a better one. Thank you

-It is not clear how the 16 studies were excluded in this step.

• Response : 14 studies reported different outcomes and study population ,and 2 study reported data inconsistency 

-(About associated factors) on page 10, are the associated factors corresponding to the last section of the results on page 20, i.e. sex and educational level?

• Response: Because we did not have enough studies to pool the OR, we presented it as a synthesis. Furthermore, we agreed that it is better for the readers if the table is included in the main text rather than uploaded as an additional file, because the findings are important. Table 4 was thus incorporated into the main text. Thanks 

-(About additional file 3 and 4). Can the authors explain why the 14 studies split into two additional files (3 and 4)? The additional file 3 is not finished.

• Response: dear reviewer, A Methodological Guidance for Systematic Reviews of Observational Epidemiological Studies Reporting Prevalence and Cumulative Incidence Data (Source: https://scholar.google.com/scholar?cluster=1187082259107005092&hl=en&as_sdt=0,5) by Munn et al. states that the Joanna Briggs Institute (JBI) has a number of tools already developed for assessing the quality of various quantitative study designs. And the critical appraisal tool used for assessing purely descriptive studies (incidence or prevalence,which assess 8 questions) should not be used for those that are analytic cross-sectional studies (which assess 9 questions). As such, we used two separate checklists. Lastly, the additional file 3 is now finished. Thank you.

-(About quality score ) Suggest to put the scores listed in the additional files on Table 1, rather than simply showing "low risk" in the last column.

• Response: Corrected, Thank you.

-( Citation )-on page 12, citations of all those technological methods such as Cochran's Q test, Higgins and Thompson’s I2 statistics and Der Simonian and Laird’s pooled effect are needed.

• Response : reviewer suggestion was appropriate, we incorporated the comment , thank you.

- page 13, "Google" not mentioned on page 8.

• Response : thank you, now it is incorporated 

- page 14, should it be n=517[32] instead of n=412[27]?

• Response: corrected ,thank you. But ,the largest sample size was 571 (Gebeyehu et al) not 517

-page 16, should it be “favorable attitudes” instead of “favorable level” in the title of figure 3?

• Response: reviewer’s suggestion is appropriate, we corrected it now . Thank you!

0 - 100% on the x axis needed in Figures 2, 3, and 4.

• Response : we corrected them and upload,thank you .

-(About subgroup meta-analysis)- page 16 & 17, suggest making a table for the section of “subgroup meta-analysis”.

• Response : the reviewer’s suggestion valid. We made additional table (table 2) for subgroup analysis comprised of the following components: outcome,participant with reference ,number of studies , total participants, pooled estimate (subtotal) with 95% confidence interval(CI),heterogeneity ,and P.value. Thank you!

-(About interpretation of funnel plots) - page 20, the authors can consider adding a caption of how to interpret/read the funnel plot in figures 5, 6, and 7. e.g. 95% CIs are represented by the two dashed lines; x and y axis represent study result/log-odds-ratios and precision/standard error; what is "log(pr)" on the x axis?

• Response: The reviewer’s suggestion is appropriate and accepted .Th has been corrected as follows: Funnel plots of publication bias.(Panel A) For good level of AMR knowledge.(Panel B) Favorable attitudes towards AMR.(Panel C) A good level of AMR practices . The Y-axis shows the standard error of the estimate (prevalence). The 95% confidence interval is represented by the dashed lines. The X-axis represents the estimate (i.e prevalence ) and the dots show the distribution of individual studies.Thank you!

-(about combine graphs)The authors can consider to combine 3 figures into one. (And combine Figures 2, 3, 4 similarly).

• Response: figures 2,3,and 4 combined in to one ; similarly figures 4,5,and 6 combined in to one .Thank you 

- page 20, additional file 5: the citation numbers [25 & 30] should be [27 & 32] in the main text.

• Response: Corrected .Thank you 

- page 27, ref #24: minor styling issue

• Response : Corrected .Thank you

Point by point response letter to reviewer Two

Greetings, dear Reviewer 2: I'm writing to express my gratitude of appreciation for general comment and taking the time to review our manuscript, "A systematic review and meta-analysis of antimicrobial resistance knowledge, attitudes, and practices: current evidence to build a strong national antimicrobial resistance narrative in Ethiopia." We thoroughly addressed specific points made and uploaded a marked-up copy (revised manuscript with tracked changes) as a "Revised Article with Changes Highlighted” file, and a clean copy as “manuscript” file. At this point, we would like to thank you for taking the time to provide invaluable comments to enrich the contents and make them more palatable to the scientific community in general.

Background

Comment: Good but it is better to rewrite the paragraphs consistently i.e., from general to specific or specific to general (world to Ethiopia or Ethiopia to the world). Example Page 5 paragraph 1 about UK, paragraph 2 about Ethiopian and the last paragraph about Europe, Pakistan and Japan.

Response: The reviewer’s suggestion about highlighting the entity under scrutiny (about AMR) from general to specific or global to local with more emphasis to study settings (Ethiopia) while reviewing important literatures is appropriate and acceptable. In the revised version, we incorporated the comments as well as re-phrasesed paragraphs ,typography of the introduction in general with specific emphasis to your comments and can be seen with marked-up manuscript .Thank you very much!

Methods and materials section

Comment: Well organized .It is better to indicate time frame /period of review 

Response: The search was double-blinded and conducted from 1st,October to 31th ,December 2022, by two authors (BW and MO). Thank you 

Comment : (Inclusion & exclusion criteria): inclusion criteria/condition, knowledge about, delete the word about

Response : deleted .Thank you 

Comment :( Context/Setting) Justify your possible reasons why you include peer reviewed journals published between January 2010 and December 2022

Response: the stud period was decided by the authors considering that using data more than 10 to 15 years might be outdated. Mellinium development goals, health sector development programs, and sustainable development programs were also taken in to consideration. Furthermore ,the AMR stewardship program in Ethiopia was in past few years and therefore studies addressing the issue are not sufficient as we did preliminary search on the policy and reports . Thank you!

Comment: Exclusion criteria: add studies not available as free full-text as exclusion criteria

Response : incorporated in to the paragraph “studies without full-text access; articles that contained insufficient information on the outcomes of the interest (knowledge, attitudes, and practices towards AMR) ; studies not available as free full-text; findings from personal opinions; articles reported outside the scope of the outcome of interest; qualitative study design; case reports; case series; letters to editors; and unpublished data were excluded!” Thank you!

Comment :( Information sources and search strategy).Ethiopians’ knowledge of, attitudes delete the word of

Response : we corrected it .thank you!

Result section 

Comment : Well organized .You have included studies in your review based on quality assessment. Results in table 1 indicated that the quality is low but quality assessment of studies using JBI’s critical appraisal tools showed the quality score is 9. It makes confusion, please make it clear

Response : Dear reviewer, we appreciate your observation. Yes! The narration creates ambiguity as a result now we put the quality score in the last column of table 1. Similarly word “ low quality” removed from the text and can be read as “Studies with a score of 7 or higher after being evaluated against these criteria were included in this SR and MA. Thank you

Comment: The funnel plot and Egger test indicated different publication bias, how to reconcile these results 

Response: Yes! On visual inspection the funnel plots were asymmetrical. We further run counter enhanced funnel plot. However, the egger test results were insignificant for KAP corroborating absence of publication bias. Dear, reviewer, according to Simmonds, M, in most systematic reviews, visual inspection of a funnel plot may give a misleading impression of the presence or absence of publication bias. Therefore, formal statistical tests for publication bias should generally be preferred (source: https://systematicreviewsjournal.biomedcentral.com/articles/10.1186/s13643-015-0004-8 )

Comment : Figure 3: Forest plot of pooled estimates for favorable level of AMR, please add attitude

Response : Incorporated .Thank you!

Comment : Why you did not indicate the KAP difference among location/ region?

Response : there was no sufficient studies in regions other than Amhara (addis Ababa,1 study,and DireDawa,1 study ) to indicate KAP difference .Rather, we considered sub group analysis by participants .Therefore, this is an important issue to indicate in the limitation of the study ,and recommend national survey .Thank you!

Comment: It is difficult to change the attitude of human on antibiotic use and AMR. In your finding the level of favorable attitude was higher than knowledge and practice. What shows this finding?

Response: dear,reviewer ,yes! In principle adequate information (knowledge ) leads to better practices and favorable attitudes. However, in primary studies involved in this study heterogeneous outcomes were reported ,meaning some studies that assessed level of knowledge did not assess attitudes simultaneously so that their denominators differ ( i.e when computesd for obvious reason ,attitudes might be higher than knowledge ).Otherwise, had it been in primary study sometimes participant might have favorable attitudes of certain condition without having knowledge about it . For instance a mother may take her child to health center for immunization because her neighbor get vaccinated her child (rare scenarios ).

Discussion section

Comment : Well organized .It is better to state the possible reasons/associated factors for significant knowledge and practice gap and favorable level of attitude regarding AMR in Ethiopia.The comparisons are quite superficial and there is little discussion of the reasons for similarity / difference to previous work. The discussion section would benefit greatly from a description of the limitations of the current work.

Response: We added more relevant articles and discussed why discrepancies in observed results in detail with respect to KAP with specific focus on participants (general population, healthcare workers, students, and animal producers) so that it indicates in whom the educational innervations should be targeted. Changes were presented with marked-up document .We accepted the comment gratefully, thank you.

Conclusion and recommendations

Your conclusion and recommendations are well organized. Since almost all studies included in the review are done in Amhara region, your data did not enough to generalized /concluded and recommend the KAP of AMR in Ethiopia. So, it is better to rewrite this part accordingly.

Response: we corrected /re-written it as “Responding to AMR threats requires policymakers,researcers, intersectoral and multidisciplinary collaboration, and local and international stakeholders to assess the population's KAP toward AMR. The pooled random effect meta-analysis in this review revealed a significant knowledge and practice gap regarding AMR among the general public, patients, and livestock producers. As a consequence,targeted educational interventions need to be in place to raise individual’s understandings about AMR and to develop effective AMR counter measures. Furthermore,the current review provides an opportunity to conduct national wide survey to address the limitations identified in the review” .Thank you

---

## [Decision Letter · Decision Letter 1]

30 May 2023

A systematic review and meta-analysis of antimicrobial resistance knowledge, attitudes, and practices: current evidence to build a strong national antimicrobial drug resistance narrative in Ethiopia

PONE-D-23-06485R1

Dear Dr. Woldegeorgis,

We’re pleased to inform you that your manuscript has been judged scientifically suitable for publication and will be formally accepted for publication once it meets all outstanding technical requirements.

Kind regards,

Mohammed Feyisso Shaka, MPH

Academic Editor

PLOS ONE

Additional Editor Comments (optional):

Reviewers' comments:

Reviewer's Responses to Questions

**Comments to the Author**

1. If the authors have adequately addressed your comments raised in a previous round of review and you feel that this manuscript is now acceptable for publication, you may indicate that here to bypass the “Comments to the Author” section, enter your conflict of interest statement in the “Confidential to Editor” section, and submit your "Accept" recommendation.

Reviewer #1: All comments have been addressed

Reviewer #2: All comments have been addressed

2. Is the manuscript technically sound, and do the data support the conclusions?

Reviewer #1: (No Response)

Reviewer #2: Yes

3. Has the statistical analysis been performed appropriately and rigorously? 

Reviewer #1: (No Response)

Reviewer #2: Yes

4. Have the authors made all data underlying the findings in their manuscript fully available?

Reviewer #1: (No Response)

Reviewer #2: Yes

5. Is the manuscript presented in an intelligible fashion and written in standard English?

Reviewer #1: (No Response)

Reviewer #2: Yes

6. Review Comments to the Author

Reviewer #1: (No Response)

Reviewer #2: The authors try to addressed my comments. I have not additional comments but the author/s should be prepare and submit the manuscript based on the author guideline of the journal.

7. PLOS authors have the option to publish the peer review history of their article (what does this mean?). If published, this will include your full peer review and any attached files.

Reviewer #1: No

Reviewer #2: **Yes: **Kindu Geta Abetie (PhD)

---

## [Editor Report · Acceptance letter]

2 Jun 2023

PONE-D-23-06485R1 

A systematic review and meta-analysis of antimicrobial resistance knowledge, attitudes, and practices: current evidence to build a strong national antimicrobial drug resistance narrative in Ethiopia 

Dear Dr. Woldegeorgis:

I'm pleased to inform you that your manuscript has been deemed suitable for publication in PLOS ONE. Congratulations! Your manuscript is now with our production department. 

Kind regards, 

on behalf of

Mr. Mohammed Feyisso Shaka 

Academic Editor

PLOS ONE